# Multiprofessional cross-site working between a level 1 and a level 3 neonatal unit: a retrospective cohort study

Theodore Dassios [1,2] Lucksini Selvadurai,[1] Ann Hickey,[1] Elizabeth Sleight,[1] Lisa Long,[3] Leonie Penna,[3] Vivette Wallen-Mitchell,[1] Ravindra Bhat,[1] Anne Greenough [4,5]

[1]Neonatal Intensive Care Unit, King's College Hospital NHS Foundation Trust, London, UK
[2]Department of Women and Children's Health, School of Life Course Sciences, Faculty of Life Sciences and Medicine, King's College London, London, UK
[3]Department of Obstetrics and Gynaecology, King's College Hospital NHS Foundation Trust, London, UK
[4]Women and Children's Health, School of Life Course Sciences, King's College London, London, UK
[5]NIHR Biomedical Research Centre at Guy's and Saint Thomas' NHS Foundation Trust and King's College, London, UK

**Correspondence to**
Dr Theodore Dassios; theodore. dassios@kcl.ac.uk

## ABSTRACT

**Objective** To assess the association of short-term neonatal outcomes with cross-site working of multiple healthcare professional teams between a level 3 and a level 1 neonatal unit.

**Design** Retrospective cohort study.

**Setting** A level 1 neonatal unit in London.

**Patients** All infants admitted to the neonatal unit, between 2010 and 2021.

**Interventions** The clinical service was rearranged in 2014 with the introduction of cross-site working between the level 1 unit and a level 3 unit of neonatal doctors, nurses and allied healthcare professionals.

**Main outcome measures** Admission of infants with a temperature less than 36°C, length of stay and time to first consultation by a senior team member.

**Results** A total of 4418 infants were admitted during the study period. The percentage of infants delivered at a gestation below 32 weeks was higher in the pre-cross-site period (8.9%) compared with the cross site period (3.6%, p<0.001). The percentage of infants with an Apgar score less than 8 at 10 min was higher in the pre-cross-site period (6.2%) compared with the cross-site period (3.4%, p=0.001). More infants were admitted with a temperature less than 36°C in the pre-cross site period (12.3%) compared with the cross site period (3.7%, p<0.001). The median (IQR) duration of time to first consultation by a senior team member was higher in the pre-cross-site period (1 (0.5–2.6) hours) compared with the cross-site period (0.5 (0.2–1.3) hours) (p<0.001). The median (IQR) length of stay was 4 (2–11) days in the pre-cross-site period and decreased to 2 (1–4) days in the cross-site period (p<0.001).

**Conclusions** Cross-site working was associated with lower rates of admission hypothermia, shorter duration of stay and earlier first senior consultation.

## INTRODUCTION

In the UK, neonatal care is stratified in three progressively more complex levels: level 1, which refers to special care baby units, level 2 or local neonatal units and the highly specialised level 3 (tertiary) or neonatal intensive care units (NICUs).[1] Neonatal care in England is currently delivered in 161 neonatal units out of which only 44 are tertiary centres with the

### WHAT IS ALREADY KNOWN ON THIS TOPIC

⇒ Birth of high-risk infants in a hospital with a tertiary neonatal unit is associated with better short-term neonatal outcomes but neonatal care cannot be confined only to tertiary neonatal units.

### WHAT THIS STUDY ADDS

⇒ Cross-site working of neonatal doctors, nurses and allied healthcare professionals between a level 3 and a level 1 neonatal unit was associated with improved short-term neonatal outcomes, such as lower rates of admission hypothermia, shorter duration of stay and earlier first senior consultation.

### HOW THIS STUDY MIGHT AFFECT RESEARCH, PRACTICE OR POLICY

⇒ Cross-site working models could be adopted more widely by the neonatal community to improve the care of infants born in hospitals without tertiary neonatal facilities.

remaining 70% being non-tertiary.[2] The allocation of a neonatal unit as tertiary or non-tertiary is based on clearly specified criteria relating to the level of clinical support that can be offered, staff seniority and training.[1 3] The tertiary units care for the sickest of infants and provide intensive care such as prolonged invasive ventilation, multiple modes of cardiorespiratory support and subspecialty services .

Neonatal complications arising in the first hours and days of life have significant consequences which may influence lifelong health and quality of life.[4] In some categories of infants, such as those born before 28 completed weeks of gestation and infants with antenatally diagnosed major congenital anomalies, antenatal planning is mandated by national guidelines to ensure transfer of women to a tertiary centre for delivery (prenatal or in utero transfers).[2] For these high-risk infants, birth in a hospital with a tertiary unit is associated with better neonatal outcomes such as lower odds

of severe brain injury and higher odds of survival without severe brain injury compared with infants born in a hospital without a tertiary NICU.[5] Birth in a tertiary unit is also associated with better outcomes in late or moderately born preterm infants.[6]

Despite the aforementioned better outcomes associated with birth in a hospital with a tertiary neonatal unit, birth is a natural process and most babies are born healthy, requiring little or no medical intervention. Neonatal care cannot be confined only to geographical areas with tertiary neonatal units and all pregnant women have a discussion around birth setting within their local area unless there are indications to deliver in a hospital with tertiary neonatal services. It is also important to note that infants at or close to term can be unexpectedly unwell and that premature birth cannot always be predicted resulting in unexpected births of sick and preterm infants in hospitals without tertiary neonatal services.[7]

Better outcomes in tertiary NICUs result from the presence of medical and surgical subspecialties, increased funding, enhanced training and sophisticated equipment. Perhaps, however, the most important factor is the highly skilled staff who work in these units: the neonatal nurses, doctors and allied health professionals such as dieticians, physiotherapists and speech and language therapists. As not all infants can be delivered in hospitals with tertiary units, neonatal outcomes could, thus, be improved by allocating enhanced resources to non-tertiary neonatal units and cross-site working of skilled neonatal professionals that work primarily in a tertiary setting and can be partly deployed to non-tertiary units.

King's College Hospital NHS Foundation Trust (KCH) has a busy surgical and medical NICU and is located in South East London. KCH acquired in 2013 a level 1 neonatal unit at the Princess Royal University Hospital (PRUH) and gradually implemented cross-site working of doctors, nurses and allied health professionals. We hypothesised that the implementation of cross-site working between a level 3 and a level 1 neonatal unit would be associated with improved short-term neonatal outcomes. Our aim was to test this hypothesis by comparing outcomes before and after the implementation of this model.

## MATERIALS AND METHODS
### Subjects and study design
A retrospective cohort study of all admissions to the Neonatal Unit at the PRUH between 1 January 2010 and 1 January 2022 was undertaken. The PRUH is located at Farnborough Common, Orpington, South East London, UK and is responsible for approximately 4000 deliveries and 350–400 neonatal admissions per year.[8] The hospital has a level 1 neonatal unit with 10 cots. Level 1 units in the UK look after infants requiring special care and are suitable for deliveries at 32 weeks of gestation and above that are considered low risk and deliveries at 30–32 weeks of gestation subject to risk assessment. Infants requiring intensive care (eg, invasive ventilation) or high

dependency care (eg, parenteral nutrition or short-term invasive ventilation) are transferred to a higher level unit.[1 3] Only infants who were born in the PRUH were included in this study; infants who were born in another hospital or whose main care was provided in another hospital and were transferred to the PRUH for continuation of care or predischarge, were not included as they were not thought to be representative of the care provided in the PRUH. If an infant had multiple admissions to the unit following referral from a local hospital or repatriation following specialist care, only the first episode was included in the analysis.

The level 3 unit of this study, was the tertiary neonatal intensive care centre of KCH NHS Trust, London, UK. KCH has a tertiary medical and surgical neonatal unit with 36 cots, with approximately 5000 deliveries and 700 neonatal admissions per year. It serves a diverse community of over 1 000 000 in South East London and is part of the London Neonatal Operational Delivery Network. Level 3 units in the UK are suitable for deliveries of infants at 27 weeks and below as well as infants that require high dependency and special care.[2 4]

### Data collected from the medical and nursing notes
Data were extracted from the BadgerNet Neonatal Electronic Patient Database (Clevermed, Edinburgh, UK). Mortality was defined as death before discharge from neonatal care.[9] The following data were collected:

#### Labour and delivery
Administration of any antenatal steroids (yes/no), cord arterial pH, cord arterial pH <7.10,[10] gestational age (weeks), birth weight (kg), sex (male/female), Apgar score at 10 min, time of admission, admission temperature (°C), admission blood glucose (mmol/L).

#### Neonatal care
Time to first consultation by a senior member of staff (Consultant, nurse in charge or senior trainee—in hours), mechanical ventilation (yes/no), pneumothorax diagnosed on chest radiography (yes/no), whole body hypothermia (cooling) for hypoxic ischaemic encephalopathy (yes/no), discharge home on supplemental oxygen (yes/no), postmenstrual age at discharge (weeks), total length of stay (days).

The birth weight z-score was calculated using the UK-WHO preterm reference chart[11] and the Microsoft Excel add-in LMSgrowth (V.2.77; www.healthforchildren.co.uk). Hypothermia was defined as an admission temperature of less than 36.5°C.[12] Hypoglycaemia was defined as a blood glucose concentration of less than 2.6 mmol/L.[13]

### Periods of implementation of cross-site working
#### Elements of the intervention
A comprehensive, gradual quality improvement programme was implemented that included up-skilling of the staff with cross-site education, training in newborn life support, simulation scenarios, regular multiprofessional case review meetings and increased levels of staffing. Individual

cases were regularly discussed for a second opinion with the senior neonatal Consultant on service at KCH which formed a separate rota of neonatal Consultants to the one at the PRUH. The neonatal team at KCH offered support by discussing individual cases and facilitated locating tertiary cots and transfer for tertiary care when required. Although not covered in detail in this report, a similar programme was undertaken by the maternity team aiming to improve maternity outcomes across both sites.

## Pre-cross-site period

The PRUH was acquired by KCH in October 2013. The first dedicated neonatal Consultant was in post in September 2014 and before this the paediatric consultants covering the PRUH unit covered the neonatal unit as well as the paediatric wards and paediatric Accident and Emergency. The clinical cover for the neonatal unit in the pre-cross-site period consisted of a consultant covering both the paediatric and neonatal side and a middle grade doctor (Registrar) covering both the paediatric and neonatal sides of the service. For purposes of data entry consistency and to reflect current neonatal practice, the start point of the pre-cross-site period was set in 2010. The pre-cross site-period, thus, consisted of the chronological years 2010–2014 (5 years).

## Wash-out period

The cross-site working model was partly introduced in 2015 with complete implementation in late 2018 when a total of six cross-site neonatal consultants were in post and formed a separate cross-site rota. In the years 2015–2018, a number of other changes were gradually introduced. Three fixed-term (1 year) neonatal consultants covering only the PRUH neonatal unit were appointed in 2016 and 2017. An additional dedicated neonatal tier of middle grade doctors (registrars) who covered only the neonatal unit (and not the paediatric ward and the emergency department) was introduced in 2017. Four-week nursing placements of the PRUH nurses to KCH were introduced in 2017 and reciprocal nursing placements of KCH nurses to the PRUH were introduced in 2018. Cross-site clinical guidelines were also gradually introduced and implemented in 2017 and 2018. The wash-out period between 2015 and 2018 was deemed essential as the aforementioned changes were introduced during this period and not all consultant posts were consistently filled. Thus, the wash-out period corresponded to the chronological years of 2015–2018 (4 years).

## Cross-site period

A complete rota of cross-site neonatal Consultants that were working across both KCH and the PRUH was fully implemented by 2019. A cross-site neonatal lead nurse for both sites was appointed in 2019. Nursing placements of the PRUH nurses to KCH were achieved with 80% of senior nurses undertaking a placement/rotation by 2019. The cross-site period, thus, consisted of the chronological years 2019–2021 (3 years). The year 2022 was not included as the data collection for this project started in April 2022. The total number of beds did not change during the study period.

## Statistical analysis

Continuous data were tested for normality with visual inspection of their distribution curves and the Kolmogorov-Smirnov test and were found to be not normally distributed. Thus, data were presented as median (IQR). The Mann-Whitney U test was used for comparisons of variables before and after the implementation of the cross-site working model. Binary variables were compared between the two periods with the chi square test. The temporal evolution of the length of stay in days, the time to first consultation by a senior member of the team in hours and the admission temperature in $^{\circ}C$ were presented in boxplots with each year presented as a separate boxplot. The length of stay was divided into quintiles and the quintile corresponding to the longer duration of stay was used as the dependent variable in a binary multivariable regression model with gestational age, birth weight z-score, admission temperature, arterial cord pH and admission year as covariates. This multivariable regression model was constructed to describe the relative contributions of the selected covariates to a longer duration of stay. The continuous parameter 'admission year' was used instead of the binary 'cross site period' to minimise type I and type II error by dichotomising a continuous variable. Multicollinearity among the independent variables in the regression analysis was assessed by examination of a correlation matrix for the independent variables. Cox proportional hazards analysis with length of stay as the outcome variable and the year of admission and gestational age as covariates, was used to examine the effect of the admission year on the length of stay.

Statistical analysis was performed using SPSS V.26.0 software.

## RESULTS

A total of 4418 infants were admitted to the neonatal unit at the PRUH during the study period. A total of 303 infants were repatriated to the PRUH having been born in another hospital and were thus excluded from the analysis. In the pre-cross-site period, 1299 infants and in the cross-site period 1052 infants were born and admitted in the neonatal unit of the PRUH. In the wash-out period, a total of 1765 infants were born and admitted in the neonatal unit (data not presented).

The characteristics and outcomes of the included infants in the two study periods are presented in table 1. The percentage of infants delivering at the PRUH at a gestation below 32 weeks was higher in the pre-cross-site period compared with the cross-site period (p<0.001). The percentage of infants with an Apgar score of less than 8 at 10 min of life, was higher in the pre-cross-site period compared with the cross site period (p=0.001). More infants were admitted with hypothermia and an admission temperature of less than 36°C in the pre-cross-site period compared with

**Table 1**  Outcomes of the admitted infants in the two study periods

| | Pre-cross-site N=1299 | Cross-site N=1052 | P value |
|---|---|---|---|
| Antenatal steroids | 451 (34.7) | 310 (29.5) | 0.127 |
| Cord arterial pH | 7.25 (7.12–7.32) | 7.27 (7.19–7.32) | 0.003 |
| Cord arterial pH<7.1 | 97 of 516 (18.8) | 57 of 592 (9.6) | <0.001 |
| Gestational age (weeks) | 37.1 (34.1–39.9) | 38.8 (35.9–40.3) | <0.001 |
| Gestation <32 weeks | 116 (8.9) | 38 (3.6) | <0.001 |
| Birth weight (kg) | 2.76 (2.01–3.45) | 3.14 (2.50–3.61) | <0.001 |
| Birth weight z score | −0.15 (-0.83–0.53) | −0.06 (-0.77–0.60) | 0.019 |
| Male sex | 773 (59.5) | 589 (56.0) | 0.217 |
| Apgar at 10 min <8 | 81 (6.2) | 36 (3.4) | 0.001 |
| Admission hypothermia | 346 (26.6) | 180 (17.1) | <0.001 |
| Admission temperature <36°C | 160 (12.3) | 39 (3.7) | <0.001 |
| Admission hypoglycaemia | 327 (25.1) | 245 (23.3) | 0.193 |
| Time to first consultation (hours) | 1.0 (0.5–2.6) | 0.5 (0.2–1.3) | <0.001 |
| Time to first consultation >4 hours | 134 (10.3) | 57 (5.4) | <0.001 |
| Pneumothorax | 18 (1.4) | 19 (1.8) | 0.528 |
| Cooling | 17 (1.3) | 14 (1.3) | 0.961 |
| Home oxygen | 129 (9.9) | 146 (13.8) | <0.001 |
| Mortality | 12 (0.9) | 1 (0.1) | <0.001 |
| Length of stay (days) | 4 (2–11) | 2 (1–4) | <0.0.001 |
| Postmenstrual age at discharge (weeks) | 38.1 (36.0–40.2) | 39.1 (36.5–40.4) | <0.001 |
| Weight at discharge | 2.66 (2.04–3.39) | 3.08 (2.42–3.58) | <0.001 |

Comparisons by Mann-Whitney U or χ2 test as appropriate.
Median (IQR) or N (%).

the cross-site period (p<0.001). Furthermore, the median duration of time to the first consultation by a member of the admitting team was higher in the pre-cross-site period compared with the cross-site period (p<0.001) with more parents receiving the first consultation/update after 4 hours from the time of admission compared with the cross-site period (p<0.001). The median length of stay was higher in the pre-cross-site period compared with the cross-site period (p<0.001). The incidence of an arterial pH of less than 7.1 was higher in the infants in whom cord gases were performed in the pre-cross-site period compared with the cross-site period (p<0.001). There were no significant differences in the administration of antenatal steroids, admission hypoglycaemia and therapeutic cooling between the two study periods (table 1). The evolution of the length of stay, time to first consultation and admission temperature over the years of the study period is presented in figure 1.

Following binary regression analysis, the highest quintile of the length of stay (10–89 days) was independently associated with the admission year (OR 0.90, 95% CI 0.86 to 0.93, adjusted p<0.001), gestational age (OR 0.72, 95% CI 0.69 to 0.75, adjusted p<0.001), birth weight z-score (OR 0.71, 95% CI 0.63 to 0.81, adjusted p<0.001), admission temperature (OR 0.18, 95% CI 1.33 to 2.31, adjusted p<0.001) and cord pH (OR 17.75, 95% CI: 5.09 to 61.81,

adjusted p<0.001). Cox proportional hazards analysis demonstrated that the admission year was significantly associated with the length of stay (HR 1.04, 95% CI 1.03 to 1.05, adjusted p<0.001) after correcting for gestational age (HR 1.17, 95% CI 1.16 to 1.18, adjusted p<0.001).

## DISCUSSION

We have demonstrated that cross-site working between a level 3 and a level 1 neonatal unit was associated with improved short-term neonatal outcomes including less admission hypothermia, shorter duration of stay and more timely first consultation by senior members of the team.

Our results highlight a successful method in which a level 1 unit can maintain and improve quality standards by working together with a tertiary unit. This has implications within networks of neonatal care, as such models could be adopted more widely to improve the care of infants born in hospitals without tertiary neonatal facilities. Of note, the results of our study relate to a unit with a workload which is higher compared with other level 1 units in London in the same study periods. A recent survey of neonatal units reported live births ranging from 2 to 3 thousands per year in other level 1 units in

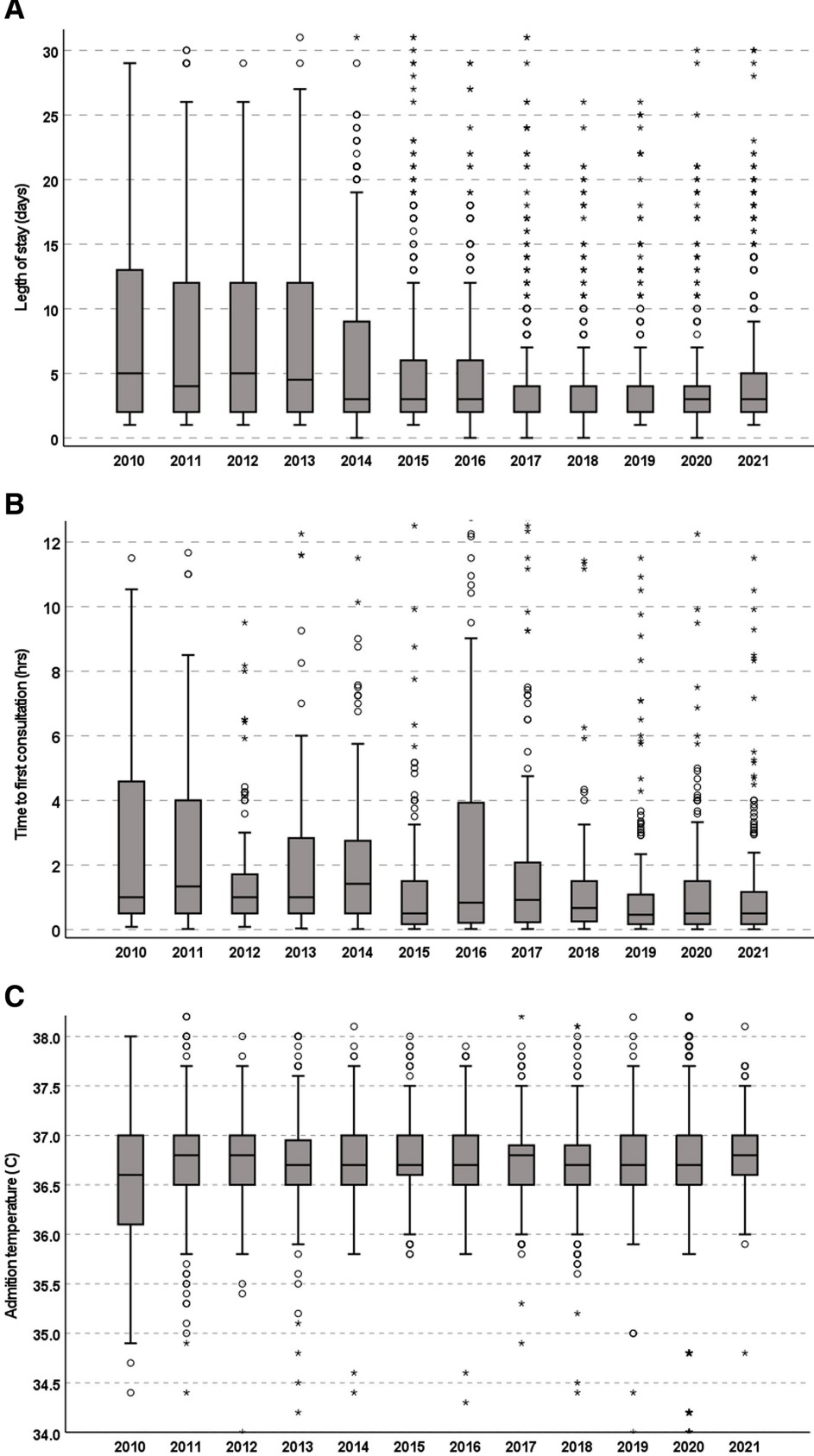

**Figure 1** Length of stay (A), time to first consultation (B) and admission temperature (C) over the study period. The horizontal lines in the boxes represent the lower quartile, median and upper quartile values of the presented parameters and the whiskers the minimum and maximum values. The whiskers do not include the outliers, defined as values more than one and a half box lengths from the median which are presented as open circles. The whiskers do not include the extreme outliers, defined as values more than three box lengths from the median which are presented as asterisks.

London.[14] Thus, our method could be feasible in the lower-volume level 1 units.

We could not compare our study results with studies reporting on interventions to improve neonatal outcomes on level 1 units, as to the best of our knowledge, there are no other such studies in the literature. Cross-site working between a level 3 and a level 1 unit is not the only potential model of care that could theoretically improve neonatal outcomes in level 1 units. Other described models include the development of the roles of the enhanced or advanced neonatal nurse practitioners, physician associates or associate specialists.[14] These, however, are mostly recent developments with a paucity of published evidence on the impact of these models on neonatal outcomes. A number of other factors can affect neonatal outcomes such as the impact of culture and morale on the workforce, investment in clinical supervision and training, implementation of quality improvement projects and integrated family development care. We could not capture these outcomes, as they are not quantified and documented in the neonatal data entry software.[14]

Some of the improved outcomes that we have reported might also be explained by improved maternity care which might be the product of cross-site working in the maternity services. Such improved outcomes include a smaller percentage of infants born with a low cord pH and a lower number of infants born at less than 32 weeks of gestation, signalling more efficient in utero transfers of women at risk. The consistent presence of neonatal consultants who did not have commitments to other paediatric services facilitated better and more frequent communication and interaction between the neonatal team and the obstetric and midwifery teams. Furthermore, improved admission temperature might also be related to better maternity care in the delivery suite, obstetric theatres and postnatal wards. The non-significant increase in the administration of antenatal steroids might be explained by the smaller percentage of infants that delivered prematurely in the PRUH and would thus qualify for this intervention.[15] The smaller percentage of infants that had a lower Apgar score at 10 min might be related to improved resuscitation at birth, but could also be partly explained by the smaller number of infants that were born with a lower cord arterial pH. We recognise, thus, that maternity improvement clearly played a major part in our improved neonatal outcomes. The majority, though, of the reported outcomes in our study are direct quality indicators of neonatal care and could be partly attributed to improved multiprofessional neonatal care.[1] We should also note that the smaller percentage of infants that delivered below 32 weeks of gestation in the cross-site period might have contributed to some improved outcomes that are more relevant to preterm infants such as admission hypothermia and a decreased length of stay.

Our study has significant financial implications. Neonatal services are costly as they require expensive equipment, consumables and employment of a skilled multiprofessional workforce which operates 24 hours a day. Neonatal units have been reported to have an incomplete understanding of the costs of running the service.[16] We did not perform a separate financial analysis, as all care days were at the same level and cost. We reported, however, that the median length of stay was halved, which corresponds to a cost reduction of a similar magnitude. It follows that if this intervention was implemented on a wider scale, it could be associated with significant cost reductions. Daily charges for neonatal care vary across different Trusts, with little consensus on the basis on which these charges are determined, but an average cost of special care per day is approximately GBP500.[17] At a single level 1 unit, thus, with 400 admissions per year the cost saving would be approximately GBP100 000 per year. We do not present in this paper a full financial analysis of the intervention but these numbers should be taken in the context of the relevant staffing and equipment expenditure. Irrespective of the financial aspects, it is important to remember that decreased length of stay and earlier discharge by senior clinicians promote normal family life in a home environment with enhanced quality of sleep, less exposure to infection risks[18] and improved neurodevelopment.[19]

Interestingly, the introduction of a cross-site model was not associated with a reduction of the total number of admissions per year. This might be explained by the fact that newborn infants constitute an already high-risk patient population. According to the WHO in 2019, 47% of all deaths in under 5 years occurred in the newborn period.[20] Previous neonatal studies have focused on improving outcomes of premature infants and more so the extremely premature ones. It is interesting, however, to note that the vast majority of births are term or late preterm and neonatal admissions are commonly infants who require a brief episode of care. Our study is thus relevant to a wider population than the results of the specialised studies of extreme prematurity and more applicable to the wider public.

Our study has strengths and some limitations. We used a large population of infants that were cared for on the same unit minimising potential differences that relate to different clinical practices or standards of care between units. The applicability of our study lies in that the improved short-term neonatal outcomes following the introduction of cross-site working, if confirmed by larger and more diverse-population studies, could strengthen the argument of implementing a cross-site neonatal model of care. We acknowledge as a limitation that it is impossible to completely separate the relative contribution of cross-site working to improved outcomes from the overall global tendency for better outcomes in neonatal care. For example, there has been an international drive to recognise and prevent neonatal hypothermia following the recognition of the detrimental effects of this complication on neonatal outcomes.[21] The International Liaison Committee on Resuscitation recommended in a consensus statement in 2015 that the admission temperature of newborn infants should be maintained between 36.5°C

and 37.5°C after birth, during stabilisation and admission, as temperature was a strong predictor of mortality and morbidity.[22] Other outcomes though, such as earlier consultation and discharge and timely in utero transfer, can be predominantly attributed to the implementation of novel workforce models such as cross-site working. Our study cannot establish a clear causal link between the introduction of cross-site working and improved neonatal outcomes as it is retrospective and spans over 12 years. Nevertheless, it is important to share with the neonatal community our experience of this intervention which could have significant policy implications.

In conclusion, cross-site working between a level 3 and a level 1 neonatal unit was associated with improved short-term neonatal outcomes relevant to level 1 neonatal care, such as a reduction in admission hypothermia, shorter length of stay and more timely first consultation by senior members of the team.

**Contributors** TD conceived the study, contributed to data analysis and interpretation, wrote the first version of the manuscript and is responsible for the overall content as the guarantor. LS collected the data and contributed to data analysis and writing the manuscript. AH led the implementation of the intervention, contributed to study design and critically revised the manuscript. ES led the implementation of the intervention and critically revised the manuscript. LL contributed to the study design and critically revised the manuscript. LP contributed to the study design and critically revised the manuscript. VW-M contributed to the study design and critically revised the manuscript. RB led the implementation of the intervention, contributed to study design and critically revised the manuscript. AG contributed to project supervision, funding acquisition and extensively revised and edited the manuscript. All authors read and agreed to the published version of the manuscript.

**Funding** This project was partially supported by King's College London 2018 Medical Research Council Confidence in Concept Award through the King's Health Partners' Research and Development Challenge Fund.

**Competing interests** None declared.

**Patient and public involvement** Patients and/or the public were not involved in the design, or conduct, or reporting, or dissemination plans of this research.

**Patient consent for publication** Not applicable.

**Ethics approval** The study was registered with the Clinical Governance Department of King's College Hospital NHS Foundation Trust. The Health Research Authority Toolkit of the National Health System, United KingdomUK confirmed that the study was not considered as research and hence would not need regulatory approval by a research ethics committee.

**Provenance and peer review** Not commissioned; externally peer reviewed.

**Data availability statement** Data are available on reasonable request.

**ORCID iDs**
Theodore Dassios http://orcid.org/0000-0001-5258-5301
Anne Greenough http://orcid.org/0000-0002-8672-5349

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
