## [Reviewer comments · BMJ Paediatrics Open]

ARTICLE DETAILS

TITLE (PROVISIONAL)	Multi-professional cross-site working between a level one and a level three neonatal unit: a retrospective cohort study
AUTHORS	Dassios, Theodore Selvadurai, Lucksini Hickey, Ann Sleight, Elizabeth Long, Lisa Penna, Leonie Wallen-Mitchell, Vivette Bhat, Ravindra Greenough, Anne

VERSION 1 – REVIEW

REVIEWER	Reviewer name: Dr. Peter Flom Institution and Country: Peter Flom Consulting, United States Competing interests: None
REVIEW RETURNED	25-Jun-2022

GENERAL COMMENTS	I mostly confine my remarks to statistical aspects of this paper. Unfortunately, I think a much better method is available. One non-statistical note: On p. 7, it might be easier to follow if the authors divided the data collected into that collected on the mother and that collected on the baby. Statistical issues: Rather than divide the time into pre-, post-, and washout, it would be better to look at time as a continuous variable. You could then use splines to look for where changes in the outcome variables happened. What was done isn't exactly *wrong*, it doesn't violate any assumptions, but categorizing a continuous variable increases type I and type II error. I wrote a blog post on this: https://medium.com/@peterflom/what-happens-when-we-categorize-an-independent-variable-in-regression-77d4c5862b6c p. 9 lines 35-36 Which data were tested for normality? You have lots of variables. Some are not even continuous. Were any of the variables normal? Also, tests for normality are not very useful as (like all tests) p values are conflated with sample size. It's never wrong to present median and IQR, but the decision to do so shouldn't be based on a KS test. lines 36-41 Rather than test each variable separately, I recommend regression with several independent variables added at once, based on substantive reasons. For admission, logistic regression should be used (assuming admission was yes/no, which seems likely). For LOS, I suggest quantile regression, looking at several quantiles. LOS in a hospital is usually right skew, and is the case here (see figure 1). Quantile regression would let you see what variables are related to any particular quantile; this seems likely to be of great interest. Figure 1 (which I looked at after making the above comments) reinforces them. The median LOS did not change that much (from
--

	about 5 days to about 3 days) but the 75th percentile changed a lot (from 12 days to 5). Table 1 - please check the result for maternal age. The values are almost identical, but the p value is 0.02. - there should be a footnote saying what test was used to get the p values
--	--

REVIEWER	Reviewer name: Dr. Paul Fleming Institution and Country: Homerton University Hospital, United Kingdom of Great Britain and Northern Ireland Competing interests: None
REVIEW RETURNED	18-Jul-2022
GENERAL COMMENTS	Dear Authors, Thank you for the opportunity to review your manuscript. You should be commended for the clarity and flow of the text. Your hypotheses are clear and you have addressed them appropriately in the text. Please see some specific comments below.  Page 4: The following may be different in an international context (it is my understanding for example that it is the other way around in Australia). Therefore, I would just clarify this statement by adding 'In the United Kingdom.....before 'Neonatal care is stratified in three progressively more complex levels: level one, which refers to special care baby units, level two or local neonatal units and the highly specialised level three (tertiary) or neonatal intensive care units (NICU). Page 6: re London Operational Network should be changed to London Neonatal Operational Delivery Network. Page 6: Can you please provide evidence of this exclusion: The Health Research Authority Toolkit of the National Health System, United Kingdom confirmed that the study was not considered as research and hence would not need regulatory approval by a research ethics committee. Page 9: Please re-word the following sentence: The data were tested for normality with the Kolmogorov-Smirnov test, were found to be not normally distributed and were thus presented as median. Care of babies at <32 weeks is reduced by almost 2/3's in the post-implementation period. Could the authors please comment on the extent to which this reduction might influence the reported outcomes. Page 13: please correct the typo here: 'Certain of the improved outcomes that we.....' I think something has gone astray with the labels attached to Figure 1 in all of the subfigures (A, B & C).

VERSION 1 – AUTHOR RESPONSE

Dr. Anne Smits, Associate Editor

Prof. Imti Choonara, Editor in Chief

BMJ Paediatrics Open

22nd July 2022

Dear Professor Choonara and Dr Smits,

Re: bmjpo-2022-001581

“Multi-professional cross-site working between a level one and a level three neonatal unit: Impact on clinical outcomes”

We thank you and the reviewers for evaluating our work and the useful comments. We are delighted to respond with a manuscript modified to answer the comments and a point by point response to each of the comments.

Yours sincerely

On behalf of the authors

Dr Theodore Dassios, King's College London

Editor in Chief Comments to Author:

Comment 1: Title replace "Impact on clinical outcomes" with "a retrospective cohort study"

Response: We have now made the replacement as suggested.

Comment 2: Be cautious in your conclusions - it is a before and after study. You overstate the relevance of your study considerably.

Response: Thank you for highlighting this point. We have now removed the word “impact” from both the title and the abstract. We also state in the discussion that “Our study cannot establish a causal link between the introduction of cross-site working and improved neonatal outcomes as it is retrospective.”

Comment 3: Discussion Page 15, lines 44-47 delete "This is, to our knowledge, the first neonatal study to assess the effect of cross-site working on neonatal outcomes" Journal policy is for authors to avoid describing their study as the first.

Response: We have now removed the phrase as suggested.

Comment 4: Respond in full to the Associate Editor and reviewers

Response: We believe we have now responded in all the comments as highlighted in the

appended responses.

Associate Editor - Comments to the Author:

I have read the paper with interest and hereby provide you with my comments:

Comment 5: Methods p 7: It is not clear to me why 'time to first consultation by a senior member of staff' was collected? What was the aim to assess this parameter? I assume that at birth of a preterm always a staff member is present?

Response: Thank you for reviewing our work and the kind comments. This parameter is collected as an indicator of how soon the parents are spoken to by a senior member of the team (Consultant, nurse in charge or senior trainee). We have now clarified in the text: "time to first consultation by a senior member of staff (Consultant, sister in charge or senior trainee - in hours)".

Comment 6: Methods p 7: 'increased levels of staffing': can the authors define what they mean with this? Which percentage of upscaling is reached? Do they mean an increased number of staff persons or an increased skills level? Please provide more information in this?

Response: The phrase "increased levels of staffing" refers to both increased number and skills level as highlighted in the "washout" and "implementation" paragraphs: "a total of six cross-site neonatal Consultants were in post and formed a separate cross-site rota." We have also clarified: "An additional dedicated neonatal tier of middle grade doctors (registrars) who covered only the neonatal unit (and not the paediatric ward and the emergency department) was introduced in 2017." Elements relating to enhanced training (simulation, cross site nursing placements) are also explained in the relevant paragraphs.

Comment 7: Methods p7: the neonatal team at KCH offered support and facilitated transfer for tertiary care when required. I was wondering how this differs for the PRUH center compared to another level 1 centers. Do children from another level 1 center do not get facilitated transfer? So the question is, in which way these 'logistics' differ for the PRUH center compared to other level 1 centers referring their patients to KCH?

Response: Thank you for highlighting this point. We completely agree that in theory all level 1 centres have access to facilitated up-transfers, however the reality is that the enhanced understanding between members of the same team is an added parameter that can facilitate

timely transfer. We have now clarified in the methods that “The neonatal team at KCH offered support by discussing individual cases and facilitated locating tertiary cots and transfer for tertiary care when required.”

Comment 8: Results Table 1: maternal age differs statistically between both periods (p0.016). However, in both periods median age is 32 years. Can the authors comment on the statistical difference, and how to interpret this?

Response: Thank you for highlighting this point. We completely agree that the clinical utility of this information is limited and have now removed this parameter.

Comment 9: Discussion: in the first sentence ‘associated with improved neonatal outcomes’: I think the authors have to attenuate this by stressing the fact that these outcomes are ‘short-term’ outcomes, and not long-term outcomes? I think this can be adapted throughout the manuscript, as this makes the context more clear to the reader.

Response: We have now altered to “short-term neonatal outcomes” in the abstract, key messages, introduction and discussion.

Comment 10: Discussion p 14/22: ‘more efficient in utero transfer of women at risk’. Can the authors support this with numbers of transfers?

Response: This was also our intention but unfortunately we could not collect this data for all periods of this study, as it was not consistently collected in the pre-implementation period.

Comment 11: Discussion p 14/22: The authors mention that improved neonatal care might in part contribute to the improved neonatal care. I miss some further reflection on this, as it might indeed impact neonatal outcomes. Can the authors further elaborate on this in the discussion?

Response: We presume that the reviewer refers to “improved maternity care might contribute to improved neonatal outcomes”. We have highlighted in the discussion that “Such improved outcomes include a smaller percentage of infants born with a low cord pH and a lower number of infants born at less than 32 weeks of gestation, signalling more efficient in utero transfers of women at risk ...Furthermore, improved admission temperature might also be related to better maternity care in the delivery suite, obstetric theatres and postnatal wards... The smaller percentage of infants that had a lower Apgar score at 10 minutes might be related to improved resuscitation at birth, but could also be partly

explained by the smaller number of infants that were born with a lower cord arterial pH. We recognise, thus, that maternity improvement clearly played a major part in our improved neonatal outcomes.”

Comment 12: Figures: please revise the legends of the separate figures, as there seem to be an error in the separate legends?

Response: Thank you for spotting this problem; we have now edited the legends to the figures.

Reviewer: 1 Comments to the Author

Dr. Peter Flom,

Comment 13: I mostly confine my remarks to statistical aspects of this paper. Unfortunately, I think a much better method is available.

Response: Thank you very much for reviewing our work and the very useful comments.

Comment 14: One non-statistical note: On p. 7, it might be easier to follow if the authors divided the data collected into that collected on the mother and that collected on the baby.

Response: Thank you for this useful suggestion. We have now divided the collected data in “Labour and delivery” and “Neonatal care”.

Comment 15: Statistical issues: Rather than divide the time into pre-, post-, and washout, it would be better to look at time as a continuous variable. You could then use splines to look for where changes in the outcome variables happened. What was done isn't exactly *wrong*, it doesn't violate any assumptions, but categorizing a continuous variable increases type I and type II error. I wrote a blog post on this:

<https://eur03.safelinks.protection.outlook.com/?url=https%3A%2F%2Fmedium.com%2F%40peterflom%2Fwhat-happens-when-we-categorize-an-independent-variable-in-regression77d4c5862b6c&data=05%7C01%7Ctheodore.dassios%40kcl.ac.uk%7C43104f00de52>

4fc8ba2208da69bd9ab6%7C8370cf1416f34c16b83c724071654356%7C0%7C0%7C637938560322510283%7CUnknown%7CTWFPbGZsb3d8eyJWljoiMC4wLjAwMDAiLCJQIjoiV2luMzliLCJBTiI6Ik1haWwiLCJXVCI6Mn0%3D%7C3000%7C%7C%7C&data=1Zg30Q7FvdtZCtSUMNHW3L2Bq9hFXgED9SaaTHkLJks%3D&reserved=0

Response: We completely agree that dichotomising a continuous variable is not ideal and some granularity will be lost by doing so. Thank you for sharing the very interesting blog

article. As our project involved multiple interventions that happened over a long period of time, we selected as a group to use the “pre and post” approach with a washout period so that the message would be easier to digest by the wider clinical neonatal community. In an effort to minimise type I and II errors and improve data management we included the “admission year” as a continuous variable in the binary regression analysis with the upper quintile of the length of stay as the dependent variable, instead of including the binary parameter “cross site period”. We have included in the methods that “The length of stay was divided into quintiles and the quintile corresponding to the longer duration of stay was used as the dependent variable in a binary multivariable regression model with gestational age, birth weight z-score, admission temperature, cord pH and admission year as covariates. This multivariable regression model was constructed to describe the relative contributions of the selected covariates to a longer duration of stay. The continuous parameter “admission year” was used instead of the binary “cross site period” to minimise type I and type II error by dichotomising a continuous variable.”

Comment 16: p. 9 lines 35-36. Which data were tested for normality? You have lots of variables. Some are not even continuous. Were any of the variables normal? Also, tests for normality are not very useful as (like all tests) p values are conflated with sample size. It's never wrong to present median and IQR, but the decision to do so shouldn't be based on a KS test.

Response: Thank you for the opportunity to clarify this point. None of the variables in our study was normally distributed other than the weight z-scores. This is not uncommon in neonatal studies as, for example in gestational age, there will be a tail of premature babies without a similar tail in post term as babies do not deliver after 42 weeks. This spills over to other outcomes as preterm infants tend to stay longer as inpatient. Also, a small fraction of the admitted infants are sick at birth and have a low cord pH but there is no corresponding group with a high pH. As part of the initial normality testing we had also done histogram inspection and we have now clarified that “Continuous data were tested for normality with visual inspection of their distribution curves and the Kolmogorov-Smirnov test”.

Comment 17: lines 36-4. Rather than test each variable separately, I recommend regression with several independent variables added at once, based on substantive reasons.

For admission, logistic regression should be used (assuming admission was yes/no, which seems likely). For LOS, I suggest quantile regression, looking at several quantiles. LOS in a hospital is usually right skew, and is the case here (see figure 1). Quantile regression would let you see what variables are related to any particular quantile; this seems likely to be of great interest.

Response: Thank you very much for this suggestion. We have now conducted a binary regression with the upper quintile of the length of stay as an output variable.

We have now included in the methods/statistics: "The length of stay was divided into quintiles and the quintile corresponding to the longer duration of stay was used as the dependent variable in a binary multivariable regression model with gestational age, birth weight z-score, admission temperature, cord pH and admission year as covariates. This multivariable regression model was constructed to describe the relative contributions of the selected covariates to a longer duration of stay."

We have also included in the results: "Following binary regression analysis, the highest quintile of the length of stay (10-89 days) was independently associated with the admission year (Odds ratio: 0.90, 95% CI: 0.86 -0.93, adjusted $p < 0.001$), gestational age (Odds ratio: 0.72, 95%CI: 0.69 – 0.75), adjusted $p < 0.001$), birth weight z-score (Odds ratio: 0.71, 95% CI: 0.63 – 0.81, adjusted $p < 0.001$), admission temperature (Odds ratio: 0.18, 95% CI: 1.33 – 2.31, adjusted $p < 0.001$) and cord pH (Odds ratio: 17.75, 95% CI: 5.09 – 61.81, adjusted $p < 0.001$)."

Thank you also for the suggestion to do a similar analysis for admission (yes/no) but unfortunately we could not perform this analysis as all our included infants have been admitted for neonatal care and there was no denominator group of all infants irrespective of admission or not.

Comment 18: Figure 1 (which I looked at after making the above comments) reinforces them. The median LOS did not change that much (from about 5 days to about 3 days) but the 75th percentile changed a lot (from 12 days to 5).

Response: We agree with this observation. Please refer to the response to the above comment.

Comment 19: Table 1 - please check the result for maternal age. The values are almost

identical, but the p value is 0.02.

Response: As in the response to comment 8, we have now removed maternal age from the table.

Comment 20: there should be a footnote saying what test was used to get the p values

Response: We have now added a footnote mentioning: "Comparisons by Mann-Whitney U or Chi square test as appropriate."

Reviewer: 2 Comments to the Author

Dr. Paul Fleming, Homerton University Hospital

Comment 21: Thank you for the opportunity to review your manuscript. You should be commended for the clarity and flow of the text. Your hypotheses are clear and you have addressed them appropriately in the text. Please see some specific comments below.

Response: Thank you very much for the kind comments.

Comment 22: Page 4: The following may be different in an international context (it is my understanding for example that it is the other way around in Australia). Therefore, I would just clarify this statement by adding 'In the United Kingdom.....before 'Neonatal care is stratified in three progressively more complex levels: level one, which refers to special care baby units, level two or local neonatal units and the highly specialised level three (tertiary) or neonatal intensive care units (NICU).

Response: We have now added this clarification as kindly suggested.

Comment 23: Page 6: re London Operational Network should be changed to London Neonatal Operational Delivery Network.

Response: We have now altered as suggested.

Comment 24: Page 6: Can you please provide evidence of this exclusion: The Health Research Authority Toolkit of the National Health System, United Kingdom confirmed that the study was not considered as research and hence would not need regulatory approval by a research ethics committee.

Response: We have now uploaded the HRA toolkit decision as supplementary material.

Comment 25: Page 9: Please re-word the following sentence: The data were tested for normality with the Kolmogorov-Smirnov test, were found to be not normally distributed and were thus presented as median.

Response: We have now broken down this sentence to two: “The data were tested for normality with the Kolmogorov-Smirnov test, and were found to be not normally distributed.

Data were thus presented as median (interquartile range).”

Comment 26: Care of babies at <32 weeks is reduced by almost 2/3's in the postimplementation period. Could the authors please comment on the extent to which this reduction might influence the reported outcomes.

Response: Thank you for this pertinent comment. We have now included in the discussion that “We should also note that the smaller percentage of infants that delivered below 32 weeks of gestation in the cross-site period might have contributed to some improved outcomes that are more relevant to preterm infants such as admission hypothermia and a decreased length of stay.”

Comment 27: Page 13: please correct the typo here: 'Certain of the improved outcomes that we.....'

Response: We have now corrected to “Some of the improved outcomes...”

Comment 28: I think something has gone astray with the labels attached to Figure 1 in all of the subfigures (A, B & C).

Response: We have now edited and re-inserted the legends to figures 1a, 1b and 1c.

VERSION 2 – REVIEW

REVIEWER	Reviewer name: Dr. Peter Flom Institution and Country: Peter Flom Consulting, United States Competing interests: None
REVIEW RETURNED	12-Aug-2022
GENERAL COMMENTS	I confine my remarks to statistical aspects of this paper. The general approach is appropriate, but I have a couple issues to resolve before I can recommend publication A general question: Was collinearity investigated? Given the covariates, it seems likely that it was present. p 9 ... In the first part of this page, the authors compare pre- and post-. At the bottom, they say they (correctly) used time as a continuous variable. So, why dichotomize it for some analysis? Also, use a spline of time to allow for modeling of nonlinear relationships, which seem likely here. p. 9 line 42 Don't divide LOS this way. You have a time to event variable, so, use a survival analysis method such as Cox proportional hazards. Peter Flom

REVIEWER	Reviewer name: Dr. Paul Fleming Institution and Country: Homerton University Hospital, United Kingdom of Great Britain and Northern Ireland Competing interests: None
REVIEW RETURNED	13-Aug-2022
GENERAL COMMENTS	Dear Authors, Thank you for addressing all the comments previously outlined and especially for providing the HRA toolkit outcome document. On re-reading your paper, there are a few very long sentences that might benefit from pruning and one or two typos (e.g. page 9 in the section 'Cross Site Period' [projected instead of project]; a few misplaced brackets in the newly added binary regression section on page 13; page 13 '[tertiary level]' should I think should be [tertiary unit];). I would simply suggest you go through the manuscript again but the correction of these can be done without the need for me to review further iterations. It has been a pleasure to review your paper on novel and collaborative approaches to cross site working that result in improved patient outcomes. I wish you all the best, Paul Fleming

VERSION 2 – AUTHOR RESPONSE

Dr. Anne Smits, Associate Editor

Prof. Imti Choonara, Editor in Chief

BMJ Paediatrics Open

26th August 2022

Dear Professor Choonara and Dr Smits,

Re: bmjpo-2022-001581

“Multi-professional cross-site working between a level one and a level three neonatal unit: Impact on clinical outcomes”

We thank the reviewers for further evaluating our work and the useful comments. We are delighted to respond with a manuscript modified to answer the comments and a point by point response to each of the comments.

Yours sincerely

On behalf of the authors

Dr Theodore Dassios, King's College London

Reviewer: 1

Dr. Peter Flom, Peter Flom Consulting

Comments to the Author

Comment 1: I confine my remarks to statistical aspects of this paper. The general approach is appropriate, but I have a couple issues to resolve before I can recommend publication

Response: Thank you very much for further reviewing our work and for the kind comments.

Comment 2: A general question: Was collinearity investigated? Given the covariates, it seems likely that it was present.

Response: Thank you for the opportunity to clarify this point. Collinearity was indeed tested and this is why parameters such as gestational age and birth weight were not both included in the model. We have now clarified in the methods that "Multi-collinearity among the independent variables in the regression analysis was assessed by examination of a correlation matrix for the independent variables."

Comment 3: p 9 ... In the first part of this page, the authors compare pre- and post-. At the bottom, they say they (correctly) used time as a continuous variable. So, why dichotomize it for some analysis? Also, use a spline of time to allow for modelling of nonlinear relationships, which seem likely here.

Response: Thank you for this comment. We completely appreciate that dichotomising a continuous variable is not methodologically advantageous. This is why we included the linear regression with year of discharge as a continuous variable, as the reviewer kindly suggested in the previous revision cycle. From a clinical point of view, however, there are two distinct periods that correspond to a period before and a period after the intervention. We believe that the readers of this paper (the clinical community) are also likely to perceive the impact of the intervention in this way, thus we selected to retain the pre/post approach, as well as the continuous analysis. In response to comment number 4, we have now also included a new Cox regression analysis of the length of stay and the year of admission, which again is inserted as a continuous variable.

Comment 4: p. 9 line 42 don't divide LOS this way. You have a time to event variable, so, use a survival analysis method such as Cox proportional hazards.

Response: Thank you for the useful suggestion to perform a Cox proportional hazards analysis. In our previous revision we divided the LOS into quintiles following such a suggestion which was provided as part of the review comments. We have now also included the Cox proportional hazards analysis: We have included in the methods: "Cox proportional hazards analysis with length of stay as the outcome variable and the year of admission and gestational age as covariates, was used to examine the effect of the admission year on the length of stay." We have also added in the results: "Cox proportional hazards analysis demonstrated that the admission year was significantly associated with the length of stay (Hazard ratio: 1.04, 95% CI: 1.03 – 1.05, adjusted $p < 0.001$) after correcting for gestational age (Hazard ratio: 1.17, 95% CI: 1.16 – 1.18, adjusted $p < 0.001$)".

Reviewer: 2

Dr. Paul Fleming, Homerton University Hospital

Comments to the Author

Comment 5: Thank you for addressing all the comments previously outlined and especially for providing the HRA toolkit outcome document.

Response: Thank you very much for kindly reviewing this revised version of our manuscript.

Comment 6: On re-reading your paper, there are a few very long sentences that might benefit from pruning and one or two typos (e.g. page 9 in the section 'Cross Site Period' [projected instead of project]; a few misplaced brackets in the newly added binary regression section on page 13; page 13 '[tertiary level]' should I think should be [tertiary unit];). I would simply suggest you go through the manuscript again but the correction of these can be done without the need for me to review further iterations.

Response: Thank you for highlighting these points, they have now been corrected.

Comment 7: It has been a pleasure to review your paper on novel and collaborative approaches to cross site working that result in improved patient outcomes.

Response: Thank you very much.